# ATTENTIVE TASK-AGNOSTIC META-LEARNING FOR FEW-SHOT TEXT CLASSIFICATION

## ABSTRACT

Current deep learning based text classification methods are limited by their ability to achieve fast learning and generalization when the data is scarce. We address this problem by integrating a meta-learning procedure that uses the knowledge learned across many tasks as an inductive bias towards better natural language understanding. Inspired by the Model-Agnostic Meta-Learning framework (MAML), we introduce the Attentive Task-Agnostic Meta-Learning (ATAML) algorithm for text classification. The proposed ATAML is designed to encourage task-agnostic representation learning by way of task-agnostic parameterization and facilitate task-specific adaptation via attention mechanisms. We provide evidence to show that the attention mechanism in ATAML has a synergistic effect on learning performance. Our experimental results reveal that, for few-shot text classification tasks, gradient-based meta-learning approaches outperform popular transfer learning methods. In comparisons with models trained from random initialization, pretrained models and meta trained MAML, our proposed ATAML method generalizes better on single-label and multi-label classification tasks in miniRCV1 and miniReuters-21578 datasets.

## 1 INTRODUCTION

Deep neural networks have shown great success in learning representations from data, but effective training of a deep neural network requires a large number of training examples and many gradient-based optimization steps. This is mainly owing to a lack of prior knowledge when solving a new task. Meta-learning or "learning to learn" (Schmidhuber, 1987; Bengio et al., 1992; Mitchell & Thrun, 1993; Vilalta & Drissi, 2002) addresses this limitation by acquiring meta-knowledge from the learning experience across many tasks. The knowledge acquired by the meta-learner provides inductive bias (Thrun, 1998) that gives rise to sample-efficient fast learning algorithms.

Although a considerable amount of research has been devoted to deep learning based meta-learning, they tend to focus on image classification and reinforcement learning. The natural language processing (NLP) related work mainly focused on language modeling while less attention has been paid to text classification. We propose a meta-learning algorithm notably designed for few-shot text classification. In contrast to popular transfer learning based text classification approaches (Howard & Ruder, 2018) that aim to fine-tune a learned representation from a different task, our meta-learning procedure is optimized to learn across a large collection of tasks with the goal of generalization from only a few examples. This enables our model to assimilate new concepts in a more principled way guided by the meta-learner.

The proposed method closely relates to Model-Agnostic Meta-Learning (MAML; see Finn et al., 2017a) that explicitly guides optimization towards adaptive representations. While MAML does not discriminate different levels of representations and adapts all parameters for a new task, we introduce Attentive Task-Agnostic Meta-Learner (ATAML) that learns task-agnostic representation while fast-adapting attention parameters to distinguish different tasks.

In effect, ATAML involves two levels of learning: representation learning that aims to obtain task-agnostic encodings of the input text in the form of a convolutional or recurrent network, and task-specific attentive learning that optimizes the attention parameters of each task for fast adaptation. Crucially, ATAML takes into account of the importance of attention in document classification and aims to encourage task-specific attentive adaptation while learning task-agnostic text representations. It is worthwhile to note that, ATAML achieves both representation and attentive learning through meta-learning; no pretraining is involved in our ATAML algorithm.

The contribution of this work is threefold: First, we propose ATAML tailed to few-shot text classification that separates task-agnostic representation learning and task-specific attentive adaptation. Moreover, we provide evidence as to how attention helps representation learning in ATAML. Although attention mechanism has been well-studied for many NLP-related tasks, we focus on the synergistic effect of attention together with task-agnostic representation learning. Our findings reveal that, when learning from a collection of tasks, task-agnostic shared representation alone is not sufficient for good generalization. More importantly, attention facilitates the discovery of shared substructures of text representations that results in better generalization.Furthermore, we introduce a smaller version of the RCV1 and Reuters-21578 dataset—miniRCV1 and miniReuters-21578—tailored to few-shot text classification, and we show that ATAML outperforms randomly initialized, pretrained and MAML-learned models.

## 2 RELATED WORK

### 2.1 FEW-SHOT TEXT CLASSIFICATION

A great body of research in NLP emphasizes on the importance of attention in a variety of tasks (Shen et al., 2018; Lin et al., 2017; Vaswani et al., 2017). These papers show that attention is able to retrieve task-specific representation across a sequence of text encodings from CNN or LSTM to obtain a task specific representation of the input. Attention could help decompose the contents of a document into "subproblems" (Parikh et al., 2016) thus producing task-specific representations; this ability to decompose text encodings also allows us to learn shared representation across tasks.

Few-shot text classification relates closely to transfer learning that aims to transfer knowledge learned from a task to a new task. They differ in that, transfer learning typically involves a small number of tasks while meta-learning aims to aggregate the knowledge learned from a number of tasks. Another difference is that, in transfer learning, we aim to directly reuse or fine-tune some existing representation, while a meta-learner is typically optimized at adapting to new tasks. Howard & Ruder (2018) proposed a transfer learning approach ULMFiT that aims to fine-tune a pretrained language for text classification. ULMFiT achives state-of-the-art performance on many text classification tasks but has not been explored under the few-shot learning setup. We use ULMFiT as one of our baselines and find fine-tuning a language model does not work well in few-shot learning.

In the context of meta-learning for few-shot text classification, previous work tend to focus on ensemble-based approaches that are not learned in an end-to-end manner. Lam & Lai (2001) proposed a regression-based approach that recommends different classification algorithms based on characteristics of the input data. Yu et al. (2018) proposed a metric learning method that first clusters different tasks and then learns cluster-dependent metric spaces. At meta-test time the model combines different metric spaces based on similarity measure with the new task. While Yu et al. (2018) represnets a document by max-pooling the phrase-level representations, we use attention mechanism to alleviate the need for different metric spaces across different tasks.

### 2.2 META-LEARNING

Previous work on deep learning based meta-learning can be summarized as: learning representations that encourage fast adaptation on new tasks (Finn et al., 2017a;b), learning universal learning procedure approximators (Hochreiter et al., 2001; Vinyals et al., 2016; Santoro et al., 2016; Mishra et al., 2017), learning to generate model parameters conditioned on training examples (Gomez & Schmidhuber, 2005; Munkhdalai & Yu, 2017; Ha et al., 2016), and learning optimization algorithms (Bengio et al., 1992; Ravi & Larochelle, 2016; Andrychowicz et al., 2016; Li & Malik, 2017). Although these methods have experimented with language modeling, none of them explored few-shot text categorization which requires global understanding of an input document.

Our work is closely related to MAML (Finn et al., 2017a) that aims to learn adaptive representations across different tasks. To form an "episode" (Vinyals et al., 2016) to optimize the meta-learner, we sample a set of tasks $\{\mathcal{D}_1, \mathcal{D}_2, ..., \mathcal{D}_S\}$ from the meta-training set $\mathscr{D}_{\text{meta-train}}$, where $\mathcal{D}_i = \{\mathcal{D}_i^{\text{train}}, \mathcal{D}_i^{\text{test}}\}$. The meta-learner performs slow learning at the meta-level across many tasks to support fast learning on new tasks. At meta-test time, we initialize our model from the meta-learned representation $\theta$, fine-tune on task $\mathcal{D}_i^{\text{train}} \sim \mathscr{D}_{\text{meta-test}}$ and evaluate on $\mathcal{D}_i^{\text{test}} \sim \mathscr{D}_{\text{meta-test}}$. Our main novelty over MAML is that, the use of task-agnostic representation learning together with task-specific attentive adaptation leads to improved discovery of text representations for few-shot adaptation.

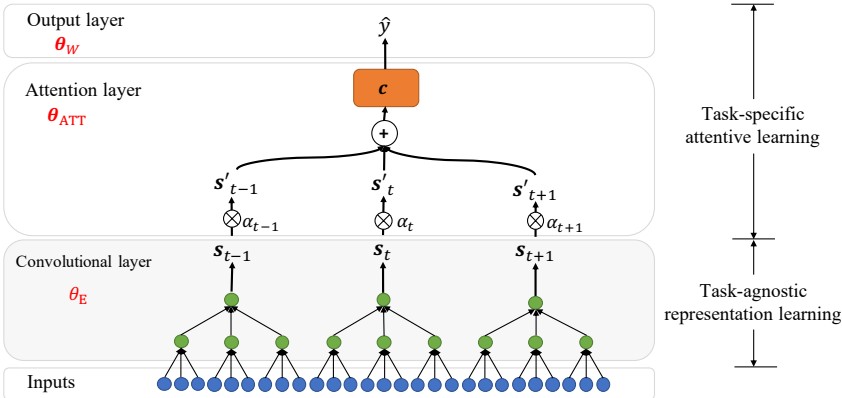

Figure 1: Network architecture of attention-based dilated convolutional network.

## 3 ATTENTIVE TASK-AGNOSTIC META-LEARNING

### 3.1 ATTENTION MODEL FOR TEXT CLASSIFICATION

As shown in Figure 1, we design an attentive neural network trained on each text classification task $\mathcal{D}$ under a loss function $\mathcal{L}$. The neural network reads the $T$-word input document $\mathbf{x} = [x_1, x_2, ..., x_T]$,

$$\mathbf{s}_t = f(x_t; \boldsymbol{\theta}_\mathrm{E}). \tag{1}$$

where $x_t$ denotes the $t$-th word. The representation learner $f(\cdot; \boldsymbol{\theta}_\mathrm{E})$ in equation 1 encodes the input sequence $\mathbf{x}$ to a corresponding sequence of states $[\mathbf{s}_1, \mathbf{s}_2, ..., \mathbf{s}_T]$, where $f$ can take the form of a recurrent or convolutional network with parameters $\boldsymbol{\theta}_\mathrm{E}$. The goal of learning $\boldsymbol{\theta}_\mathrm{E}$ in ATAML is to obtain *meta-learned* task-agnostic parameters that can provide meaningful encodings of the input text.

We then apply content-based attention mechanism (Bahdanau et al., 2014; Hermann et al., 2015; Graves et al., 2014; Sukhbaatar et al., 2015) that enables the model to focus on different aspects of the document. The specific attention formulation used here is defined in equation 2 and belongs to a type of feedforward attention (Raffel & Ellis, 2015),

$$\alpha_t = \boldsymbol{\theta}_\mathrm{ATT}^\mathsf{T} \mathbf{s}_t, \qquad \mathbf{s}_t' = \alpha_t \mathbf{s}_t, \qquad \mathbf{c} = \frac{1}{T} \sum_{t=1}^{T} \mathbf{s}_t', \tag{2}$$

where $\boldsymbol{\theta}_\mathrm{ATT}$ represents the attention parameter vector. For each memory state $\mathbf{s}_t$, we calculate its inner product with the attention parameter, resulting in a scalar $\alpha_t$. The scalar $\alpha_t$ rescales each state $\mathbf{s}_t$ into $\mathbf{s}_t'$, which are averaged to obtain the final representation $\mathbf{c}$ of a document. The attention retrieves relevant information from a document and offers interpretability into the model behavior by explaining the importance of each word, through attention weight $\alpha_t$, that contributes to the final prediction.

Once an input document $\mathbf{x}$ is encoded into the vectorized representation $\mathbf{c}$, we apply a softmax classifier parameterized by $\boldsymbol{\theta}_W$ to obtain the predictions $\hat{y}$. The softmax classifier is replaced by a set of sigmoid classifiers if the labels are not mutually exclusive in multi-label classification,

$$\hat{y} = \mathrm{softmax}(\mathbf{c}; \boldsymbol{\theta}_W) \quad \text{or} \quad \hat{y} = \mathrm{sigmoid}(\mathbf{c}; \boldsymbol{\theta}_W). \tag{3}$$

### 3.2 THE ATTENTIVE TASK-AGNOSTIC META-LEARNER

ATAML learns to obtain common representations that can be shared across different tasks while having the fast learning ability to quickly adapt to new tasks. In contrast with MAML which does not make any distinction between different parameters in the meta-learner, the proposed ATAML splits all parameters $\theta$ into two disjoint sets, shared task-agnostic parameters $\theta_\mathrm{E}$ and attentive task-specific parameters $\theta_\mathrm{T}$, and employs discriminative strategies in the meta-training and meta-testing phrases. The shared parameters $\theta_\mathrm{E}$, as shown in shaded area in Figure 1, are aimed at representation learning while the task-specific parameters $\theta_\mathrm{T}$ are aimed at capturing task-specific information for classification.

---

**Algorithm 1** Attentive Task-Agnostic Meta-Learner

---

**Require:** $\mathscr{D}_{\mathrm{meta-train}}$: the meta-train set
**Require:** $N$-way $K$-shot learning
**Require:** $S$ classification tasks for each training episode
**Require:** $\beta_{\mathrm{T}}, \beta_{\mathrm{M}}$: task and meta level learning rate
**Require:** $\theta_{\mathrm{E}}$: shared parameters for representation learning
**Require:** $\theta_{\mathrm{T}} = \{\boldsymbol{\theta}_W, \boldsymbol{\theta}_{\mathrm{ATT}}\}$: parameters to be adapted at the task level
1: randomly initialize $\theta_{\mathrm{E}}$ and $\theta_{\mathrm{T}}$           ▷ Initialize all parameters
2: **while** not done **do**
3:    Sample $S$ tasks: $\mathcal{D}_i \sim \mathscr{D}_{\mathrm{meta-train}}$      ▷ Sample tasks for meta-training
4:    **for all** $\mathcal{D}_i$ **do**
5:      $\theta'_{\mathrm{T},i} = \theta_{\mathrm{T}} - \beta_{\mathrm{T}} \nabla_{\theta_{\mathrm{T}}} \mathcal{L}(\mathcal{D}_i^{\mathrm{train}}; \{\theta_{\mathrm{T}}, \theta_E\})$     ▷ Get task-specific parameters
6:    $\mathcal{L}_{\mathrm{meta}} = \sum_{\mathcal{D}_i} \mathcal{L}(\mathcal{D}_i^{\mathrm{test}}; \{\theta'_{\mathrm{T},i}, \theta_{\mathrm{E}}\})$     ▷ Get loss of the meta-learner
7:    $\theta_{\mathrm{T}} \leftarrow \theta_{\mathrm{T}} - \beta_{\mathrm{M}} \nabla_{\theta_{\mathrm{T}}} \mathcal{L}_{\mathrm{meta}}$       ▷ Update task-specific parameters
8:    $\theta_{\mathrm{E}} \leftarrow \theta_{\mathrm{E}} - \beta_{\mathrm{M}} \nabla_{\theta_{\mathrm{E}}} \mathcal{L}_{\mathrm{meta}}$       ▷ Update shared parameters

---

### 3.2.1 META TRAINING

The Attentive Task-Agnostic Meta-Learning training algorithm is described in Algorithm 1. We use $\theta$ to denote all parameters of the model ($\theta = \{\boldsymbol{\theta}_W, \boldsymbol{\theta}_{\mathrm{ATT}}, \boldsymbol{\theta}_{\mathrm{E}}\}$), which is divided into shared parameters $\boldsymbol{\theta}_{\mathrm{E}}$ and task-specific parameters $\theta_{\mathrm{T}}$, where $\theta_{\mathrm{T}} = \{\boldsymbol{\theta}_W, \boldsymbol{\theta}_{\mathrm{ATT}}\}$.

To create one meta-training "episode" (Vinyals et al., 2016), we sample $S$ tasks from $\mathscr{D}_{\mathrm{meta-train}}$ and optimize the model towards fast learning across all sampled tasks $[\mathcal{D}_1, \mathcal{D}_2, ..., \mathcal{D}_S]$. As we are sampling random tasks from $\mathscr{D}_{\mathrm{meta-train}}$ in each meta-training iteration, the goal of the meta-learner is to obtain task-agnostic representation $\theta_{\mathrm{E}}$ that is reusable for different tasks.

For every task $\mathcal{D}_i$ in the meta-training iteration, we only update the task-specific parameters that are initialized with $\theta_{\mathrm{T}}$ and updated to $\theta'_{\mathrm{T},i}$ using task-specific gradients $\nabla_{\theta_{\mathrm{T}}} \mathcal{L}(\mathcal{D}_i^{\mathrm{train}}; \{\theta_{\mathrm{T}}, \theta_E\})$. We further calculate the expected loss across all tasks according to the post-update parameters that is composed of the task-specific fast weights $\theta'_{\mathrm{T},i}$ and shared slow weights $\boldsymbol{\theta}_{\mathrm{E}}$,

$$\mathcal{L}_{\mathrm{meta}} = \sum_{\mathcal{D}_i} \mathcal{L}(\mathcal{D}_i^{\mathrm{test}}; \{\theta'_{\mathrm{T},i}, \boldsymbol{\theta}_{\mathrm{E}}\}), \tag{4}$$

where $\mathcal{L}_{\mathrm{meta}}$ can be understood as the loss of the meta-learner. More intuitively, $\mathcal{L}_{\mathrm{meta}}$ gives us an evaluation measure on how well the task-specific parameters $\theta_{\mathrm{T}}$ can adapt across all the sampled tasks $\mathcal{D}_i$, together with a measure on how well the shared parameters $\boldsymbol{\theta}_{\mathrm{E}}$ can be reused across all tasks. The meta-optimization therefore consists of minimizing $\mathcal{L}_{\mathrm{meta}}$ with respect to all parameters $\theta$ towards optimizing the model's adaptability and re-usability across different tasks. The meta-training iterations are repeated until the model converges, and the resulting parameters $\theta$ are then used as initialization at meta-test time.

### 3.2.2 META TESTING

Meta testing involves evaluating on the meta-learned model on the meta-test set $\mathscr{D}_{\mathrm{meta-test}}$ by fine-tuning on $\mathcal{D}_i^{\mathrm{train}}$ and test on $\mathcal{D}_i^{\mathrm{test}}$, where $\mathcal{D}_i \sim \mathscr{D}_{\mathrm{meta-test}}$. We introduce a meta testing approach that freezes the shared representation learning parameters $\boldsymbol{\theta}_{\mathrm{E}}$ and only applies gradient on the task-specific parameters $\boldsymbol{\theta}_{\mathrm{T}}$. In contrast to fine-tuning all parameters for a new task, our approach provides regularization to few-shot learning that improves generalization. For the avoidance of misunderstanding, we note that labels in meta-train and meta-test sets are mutually exclusive.

### 3.2.3 GRADIENT PROPERTIES

We now draw connections between task-agnostic representation learning and task-specific attentive classification to highlight the impact of attention. Through gradient analysis in Appendix A, we find that the shared task-agnostic representation layer makes more effective gradient updates if there is a stronger match between attention $\boldsymbol{\theta}_{\mathrm{ATT}}$ and the representation state $\mathbf{s}_t$. This enables the model to focus on different aspects of the representation, and selectively updates parameters that have greater

contribution to the classification outcome. This results in an effective task-agnostic representation adept at extracting meaningful substructures from the input text.

# 4 EXPERIMENTS

We provide three sets of empirical evaluations on the single-label miniRCV1, multilabel miniRCV1 and miniRCV1miniReuters-21578 datasets to analyze the proposed meta-learning framework.

## 4.1 NETWORK ARCHITECTURE

We use Temporal Convolutional Networks (TCN), which is a type of dilated convolution (Van Den Oord et al., 2016), as our network architecture. We have also conducted experiments with bidirectional LSTM (Schuster & Paliwal, 1997) detailed in the Appendix.

The TCN contains two layers of dilated causal convolutions with filter size 3 and dilation rate 3. Each convolutional layer is followed by a Leaky Rectified Linear Unit (Maas et al., 2013) with negative slope rate 0.01, which is followed by 50% dropout (Srivastava et al., 2014). For word representation, we use 300 dimensional Glove embeddings (Pennington et al., 2014). For optimization, we use Adam optimizer (Kingma & Ba, 2014). For the loss function, we use categorical cross entropy error when each document contains only one label and sigmoid cross entropy error when each document may contain multiple labels. Although it is common to use threshold calibration algorithms for multilabel classification, we use the constant 0.5 as prediction threshold in order to reduce the impact of external algorithms.

## 4.2 DATA

Reuters Corpus Volume I (RCV1) is an archive of news stories for research on text categorization (Lewis et al., 2004). We create two versions of the miniRCV1 dataset by selecting a subset from the full RCV1 dataset to study the effect of few-shot learning in text classification:

1. *miniRCV1 for single-label classification* consisting of the 55 second-level topics as target classes. We sample 20 documents from each class which is further divided into a training set that contains 5 documents and a testing set that contains 15 documents. Documents with overlapping topics are removed to ensure each document contains a single label.

2. *miniRCV1 for multi-label classification* consisting of 102 out of 103 non-mutually exclusive labels. Each document is associated with a set of labels and we exclude one label that only appeared once in the corpus. We sample about 20 documents for each class and divide them into training and testing sets in a similar manner. It is worthwhile to mention that, due to the inherent properties of multi-labeled data (Zhang & Zhou, 2014), some classes may contain more examples than others classes.

Similar to miniRCV1, we create a smaller version of the Reuters-21578 dataset by selecting about 20 examples for each label.

## 4.3 FEW-SHOT LEARNING SETUP

At the meta-level, we divide all classes into mutually exclusive meta-train, meta-validation and meta-test sets. In the $N$-way $K$-shot setup, during meta-training, we randomly sample $N$ classes among the meta-training set where each class contains $K$ training examples. At meta-test time, we randomly sample $N$ classes among the meta-test set and calculate evaluation statistics across many runs. We evaluate 5-way 1-shot, 5-way 5-shot, 10-way 1-shot and 10-way 5-shot learning for both single-label and multi-label classification. The single-label classification task is evaluated on classification accuracy; the multi-label classification task is evaluated on micro and macro F1-scores, which are intended to measure the average F1-scores across all labels. They differ in that, micro-average gives equal weights to each example regardless of label imbalance, whereas macro-average treats different labels equally.

## 4.4 RESULTS AND DISCUSSION

As with other meta-learning paradigms we consider two baselines: models trained from random initialization, i.e., "random", and models pretrained across many sampled meta-train tasks, i.e.,

Table 1: Comparing single-label classification accuracies between baselines and ATAML on miniRCV1

| Method | | 5-way Accuracy | | 10-way Accuracy | |
| Meta | Base | 1-shot | 5-shot | 1-shot | 5-shot |
|---|---|---|---|---|---|
| random | TCN (A) | 41.52% | 65.64% | 28.32% | 45.12% |
| pretrained-1 | TCN (A) | 24.06% | 57.08% | 18.60% | 45.85% |
| pretrained-2 | ULMFiT (Howard & Ruder, 2018) | 28.46% | 61.33% | 14.72% | 60.03% |
| MAML | TCN (A) | 47.09% | **72.65%** | 31.57% | **62.75%** |
| ATAML | TCN (A) | **54.05%** | **72.79%** | **39.48%** | 61.74% |

Table 2: Comparing multi-label classification outcomes between baselines and ATAML on miniRCV1

| Method | | 5-way Micro-F1 | | 10-way Micro-F1 | | 5-way Macro-F1 | | 10-way Macro-F1 | |
| Meta | Base | 1-shot | 5-shot | 1-shot | 5-shot | 1-shot | 5-shot | 1-shot | 5-shot |
|---|---|---|---|---|---|---|---|---|---|
| random | TCN (A) | 38.9% | 60.9% | 40.6% | 45.6% | 31.4% | 55.7% | 22.9% | 33.1% |
| pretrained | TCN (A) | 26.9% | 55.8% | 33.5% | 52.1% | 17.0% | 51.5% | 14.9% | 41.4% |
| MAML | TCN (A) | 52.3% | **69.1%** | 44.9% | 58.6% | 43.2% | **64.3%** | 27.7% | **48.4%** |
| ATAML | TCN (A) | **59.7%** | **71.1%** | **50.7%** | **61.3%** | **54.3%** | **65.0%** | **38.5%** | **49.2%** |

"pretrained". In addition, we also compare our proposed ATAML framework with MAML under similar architecture. Our experiments show that while MAML achieves better accuracies compared to the aforementioned baselines, ATMAL significantly outperforms MAML in all 1-shot learning experiments. Table 1, Table 2 and Table 3 summarize these results on single-label miniRCV1, multi-label miniRCV1 and multi-label miniReuters-21578 experiments, wherein "Meta" denotes the type of meta learner, "Base" denotes the architecture of the network, "random" denotes models trained from random initialization, "(A)" denotes models trained with attention and the bold numbers highlight the best performing ones at 95% confidence interval.

**The difficulty of learning from scratch.** Few-shot text classification is a challenging task as text data contain rich information from various aspects which are difficult to ascertain from a few training examples. This difficulty is manifested in our results with the poor testing performance when trained from random initialization. Meanwhile, in both single-label and multi-label classification tasks, the TCN models with random initialization, improves significantly when the training examples are increased from 1 to 5. Furthermore, we show in the Appendix that, classic machine learning algorithms, such as support vector machine, naive Bayes multinomial and K-nearest neighbors, as well as document embedding algorithms, such as doc2vec (Levine & Haus, 1985) and doc2vecC (Chen, 2017), also suffer from data scarcity in few-shot learning. This hints at the need for effective few-shot text classification algorithms.

Why does pretrained 10-way $K$-shot TCN models perform so poorly? In multi-label classification tasks, some labels appear less frequently in the training data. This label imbalance causes uncalibrated output probabilities when using the constant 0.5 as prediction threshold. Some pretrained models performs worse than random guesses because its output probabilities are not well distributed.

**Pretrained models in few-shot learning.** In Table 1, we listed two pretrained baselines: "pretrain-1" from a collection of few-shot tasks as in (Finn et al., 2017a) and "pretrain-2" from language model ULMFiT (Howard & Ruder, 2018). "pretrain-1" performs worse than models trained from random initialization. As each task contains a small number of examples, when we pretrain the model from many tasks in the meta-training set, the sampled tasks provide contradictory supervisory signals to the classifier, hence making it difficult to pretrain effectively (Finn et al., 2017a). As for "pretrain-2" (ULMFiT), the model fails to fine-tune on 1-shot tasks. ULMFiT first fine-tunes a pretrained language model on the new dataset, then adds a classifier on top of the language model and fine-tunes the whole model for classification. This is challenging in the few-shot setup because a few-shot task only contains a small vocabulary which makes it easy to overfit the language model. We also observe that ULMFiT works better than "random" and "pretrain-1" in 10-way 5-shot where more training data is available.

Table 3: Comparing multi-label classification between baselines and ATAML on miniReuters-21578

| Method | | 5-way Micro-F1 | | 10-way Micro-F1 | | 5-way Macro-F1 | | 10-way Macro-F1 | |
|---|---|---|---|---|---|---|---|---|---|
| Meta | Base | 1-shot | 5-shot | 1-shot | 5-shot | 1-shot | 5-shot | 1-shot | 5-shot |
| random | TCN (A) | 38.2% | 66.0% | 25.1% | 44.9% | 30.6% | 55.0% | 17.9% | 33.6% |
| pretrained | TCN (A) | 23.5% | 50.3% | 18.4% | 49.1% | 16.4% | 37.8% | 12.0% | 37.3% |
| MAML | TCN (A) | 52.4% | **74.1%** | 38.1% | **61.2%** | 44.3% | 64.3% | 29.9% | **51.2%** |
| ATAML | TCN (A) | **66.3%** | **76.5%** | **42.6%** | 60.8% | **60.9%** | **69.4%** | **34.9%** | **51.2%** |

Table 4: Ablation studies on miniReuters-21578 for multi-label classification

| Method | | 5-way Micro-F1 | | 10-way Micro-F1 | | 5-way Macro-F1 | | 10-way Macro-F1 | |
|---|---|---|---|---|---|---|---|---|---|
| Meta | Base | 1-shot | 5-shot | 1-shot | 5-shot | 1-shot | 5-shot | 1-shot | 5-shot |
| random | E (A) | 36.7% | 66.1% | 25.2% | 49.1% | 29.2% | 55.0% | 18.2% | 36.8% |
| MAML | E (A) | 44.9% | 72.3% | 26.4% | 59.2% | 35.6% | 61.7% | 19.6% | 47.4% |
| MAML | TCN | 26.4% | 65.7% | 11.4% | 44.5% | 19.1% | 52.7% | 7.6% | 31.2% |
| MAML | TCN (A) | 52.4% | **74.1%** | 38.1% | **61.2%** | 44.3% | **64.3%** | 29.9% | **51.2%** |
| TAML | TCN | 21.5% | 55.7% | 11.5% | 32.1% | 15.1% | 41.5% | 7.3% | 23.7% |
| ATAML | TCN (A) | **66.3%** | **76.5%** | 42.6% | **60.8%** | **60.9%** | **69.4%** | 34.9% | **51.2%** |
| ATAML | TCN (A̲) | 62.7% | **77.5%** | **49.5%** | **63.7%** | 58.3% | **71.1%** | **41.6%** | **54.2%** |

**The effect of meta learning.** From all three experiments, the empirical results demonstrate the basic MAML with attention mechanism learners performs notably better than the non-meta-learned baselines. More importantly, the proposed ATAML algorithm offers further improvements that are statistically significant in all the 1-shot learning experiments. These empirical findings support the need for meta-learning in few-shot text classification. That being the case, the empirical findings further support the importance of learning task-agnostic representations together with task-specific attentive adaptations. To better understand the representation learning procedure as well as the role of attention in meta training, we undertake ablation studies to provide further insights into ATAML.

## 4.5 ABLATION STUDIES

### 4.5.1 THE SYNERGISTIC EFFECT OF ATTENTION AND TASK-AGNOSTIC REPRESENTATIONS

The notable feature of ATAML is the use of attention mechanism together with shared task-agnostic representations. For the avoidance of misunderstanding, we note that the shared task-agnostic representation is learned through meta-learning, which is different from the pretrained baseline methods. To show the synergistic effect of attention on the meta-learner, we construct an ablation experiment in Table 4 "TAML, TCN" that trains shared task-agnostic representation without the use of attention. The performance of "TAML, TCN" is drastically worse than all other methods, suggesting learning task-agnostic representation alone, without the use of attention, does not work well for few-shot text classification tasks. We also observe that, among all attentive models, the proposed ATAML works the best. This supports our claim that the interaction between the attentive task-specific classifier and task-agnostic representation learner facilitates learning when utilized together.

### 4.5.2 THE NEED TO LEARN STRUCTURED REPRESENTATION

With ablation studies we can offer evidence into the need to learn text in a structured manner as opposed to making classifications at the word level alone. We use "E (A)" to denote a model where an attention model is directly applied to the word embeddings. The goal of this model is to extract individual words to make predictions. This model provides a measure on classification performance if we only take into account individual word-level representations. The empirical results in Table 4 suggest classifying from word embeddings is inferior to the proposed ATAML model, indicating the need to learn text structures, such as phrase or sentence level representations. Moreover, learning from only a few examples exacer-

syria says ready to resume peace talks with israel. syria said on tuesday it was ready to resume peace talks with israel in washington from the point where they broke off in march after a wave of islamic suicide bombings in israel. foreign minister farouq al shara said some progress had been made in talks with the former labour led government of prime minister shimon peres regarding theprinciple of land for peace and security arrangements there are points which were not agreed upon and there are points which were agreed upon and the united states as a sponsor of the talks knows what was agreed upon and what was not agreed upon. shara said the talks should not start from zero point we said that syria is ready to resume the talks from the point where they stopped. he toldreporters at damascus airport as egyptian foreign minister amr moussa ended a trip to syria. moussa said he had good talks with president hafez al assad and that syria was ready to resume negotiations with israel within the framework of united nations.

Figure 2: Visualizing attentions learned by MAML TCN(A).

syria says ready to resume peace talks with israel. syria said on tuesday it was ready to resume peace talks with israel in washington from the point where they broke off in march after a wave of islamic suicide bombings in israel. foreign minister farouq al shara said some progress had been made in talks with the former labour led government of prime minister shimon peres regarding theprinciple of land for peace and security arrangements there are points which were not agreed upon and there are points which were agreed upon and the united states as a sponsor of the talks knows what was agreed upon and what was not agreed upon. shara said the talks should not start from zero point we said that syria is ready to resume the talks from the point where they stopped. he toldreporters at damascus airport as egyptian foreign minister amr moussa ended a trip to syria. moussa said he had good talks with president hafez al assad and that syria was ready to resume negotiations with israel within the framework of united nations.

Figure 3: Visualizing attentions learned by ATAML TCN(A).

bates the effect of over-fitting as it is more likely to have spurious correlations at the word level compared with phrase or sentence level. It is therefore desirable to have the ability to learn text structures.

### 4.5.3 THE ROLE OF ATTENTION IN META TRAINING

To analyze the role of attention in meta training, we construct an attention-based meta training strategy where the attention parameters are not updated in each meta training iteration. Although the attention parameters are not being updated in meta training, they take task-specific fast weights as regular ATAML during meta-testing and these fast weights have direct influence over the gradients of the TCN layers. The goal of this model is to exploit the fast weights of the attention parameters and examine if this could produce well trained representations. This model, denoted as "TCN($\underline{A}$)", has similar performance with the regular ATAML models in Table 4. Thus, the role of attention in meta training is to facilitate the learning of shared representations. This also suggests that the attention parameters are flexible in taking different directions for fast adaptation when trained on different tasks.

### 4.6 VISUALIZING LEARNED ATTENTIONS

Figure 2 and Figure 3 illustrate the the same training example after the meta-learner is trained with MAML and ATAML, respectively. The target label is "INTERNATIONAL RELATIONS" and both models make correct predictions for this training example. Whereas the MAML model illustrated in Figure 2 is over-fitting to the keyword "president", the proposed ATAML model in Figure 3 identifies multiple key phrases, such as "talk with", "agreed upon" and "negotiation with", that are important to the classification of "INTERNATIONAL RELATIONS". Learning meaningful phrase-level representations regularizes a model from over-fitting to spurious correlation in the training examples.

## 5 CONCLUSION

We propose a meta learning approach that enables the development of text classification models from only a few training examples. The proposed ATAML is designed to encourage task-agnostic representation learning by way of task-agnostic parameterization and facilitate task-specific adaptation via attention mechanisms. The use of attention mechanism is capable of decomposing some text into substructures for task-specific adaptation. Our empirical studies reveal that attention brings synergistic effect on meta-learning shared text representations. The effectiveness of the proposed meta-learning algorithm for few-shot text classification is clearly supported by our empirical studies on the miniRCV1 and miniReuters-21578 datasets. We also provided ablation analysis and visualization to get insights into how different components of the model work together.

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

# Appendices

## A GRADIENT PROPERTIES

For a standard neural network $\boldsymbol{\theta}_E$ without attention, the derivative of the loss $\mathcal{L}$ with respect to $\boldsymbol{\theta}_E$ is

$$\frac{\partial \mathcal{L}}{\partial \boldsymbol{\theta}_E} = \frac{\partial \mathcal{L}}{\partial \hat{y}} \frac{1}{T} \sum_{t=1}^{T} \frac{\partial \hat{y}}{\partial \mathbf{s}_t} \frac{\partial \mathbf{s}_t}{\partial \boldsymbol{\theta}_E} = \frac{\partial \mathcal{L}}{\partial \hat{y}} \frac{\boldsymbol{\theta}_E^{\mathsf{T}}}{T} \sum_{t=1}^{T} \frac{\partial \mathbf{s}_t}{\partial \boldsymbol{\theta}_E}. \tag{5}$$

In contrast, for an attentive neural network, the derivative of the loss $\mathcal{L}_{\text{ATT}}$ with respect to $\boldsymbol{\theta}_E$ is

$$\frac{\partial \mathcal{L}_{\text{ATT}}}{\partial \boldsymbol{\theta}_E} = \frac{\partial \mathcal{L}}{\partial \hat{y}} \frac{1}{T} \sum_{t=1}^{T} \frac{\partial \hat{y}}{\partial \mathbf{s}'_t} \frac{\partial \mathbf{s}'_t}{\partial \mathbf{s}_t} \frac{\partial \mathbf{s}_t}{\partial \boldsymbol{\theta}_E}, \tag{6}$$

where $\mathbf{s}'$ is defined by the attention mechanism in equation 2. Accordingly, we have

$$\frac{\partial \mathbf{s}'_t}{\partial \mathbf{s}_t} = \frac{\partial}{\partial \mathbf{s}_t}((\boldsymbol{\theta}_{\text{ATT}}^{\mathsf{T}} \mathbf{s}_t)\mathbf{s}_t) = \mathbf{s}_t \boldsymbol{\theta}_{\text{ATT}}^{\mathsf{T}} + \mathbf{s}_t^{\mathsf{T}} \boldsymbol{\theta}_{\text{ATT}} I, \tag{7}$$

with $I$ the identity matrix. The detailed steps to obtain equation 7 is included in Section A.1. We further rewrite equation 6 into:

$$\frac{\partial \mathcal{L}_{\text{ATT}}}{\partial \boldsymbol{\theta}_E} = \frac{\partial \mathcal{L}}{\partial \hat{y}} \frac{\boldsymbol{\theta}_E^{\mathsf{T}}}{T} \sum_{t=1}^{T} (\mathbf{s}_t \boldsymbol{\theta}_{\text{ATT}}^{\mathsf{T}} + \mathbf{s}_t^{\mathsf{T}} \boldsymbol{\theta}_{\text{ATT}} I) \frac{\partial \mathbf{s}_t}{\partial \boldsymbol{\theta}_E}. \tag{8}$$

By comparing the derivatives of the standard and attentive neural networks, i.e., equation 5 and equation 8, we find their only difference to be in the scaling factor for each state. The gradients are scaled by $(\mathbf{s}_t \boldsymbol{\theta}_{\text{ATT}}^{\mathsf{T}} + \mathbf{s}_t^{\mathsf{T}} \boldsymbol{\theta}_{\text{ATT}} I)$ for each state $\mathbf{s}_t$ after we introduce attention. In other words, the gradients of attentive model have an additional parameterization through the interactions between the recurrent states $\mathbf{s}_t$ and the attention parameters $\boldsymbol{\theta}_{\text{ATT}}$.

This produces more expressive gradients where the updates of the shared representation not only depend on the updates of $\boldsymbol{\theta}_E$ and $\frac{\partial \mathcal{L}}{\partial \hat{y}}$, but also controlled by the attention mechanism. More specifically, if we focus on the scaling effects of the transformation, especially the diagonal matrix $\mathbf{s}_t^{\mathsf{T}} \boldsymbol{\theta}_{\text{ATT}} I$, we find the learning is more discriminative based on the similarity between cell state and attention vector. Consequently, the model makes more gradient updates if there is a stronger match between attention $\boldsymbol{\theta}_{\text{ATT}}$ and the representation state $\mathbf{s}_t$. Summing up, attention not only enables the model to focus on different aspects of the representation states, it also results in a more effective learning procedure that allows fast adaptation and generalization.

## A.1 JACOBIAN CALCULATION

Here we show the detailed steps to obtain the following Jacobian:

$$\frac{\partial \mathbf{s}'_t}{\partial \mathbf{s}_t} = \frac{\partial}{\partial \mathbf{s}_t}((\boldsymbol{\theta}_{\text{ATT}}^{\mathsf{T}} \mathbf{s}_t)\mathbf{s}_t) = \mathbf{s}_t \boldsymbol{\theta}_{\text{ATT}}^{\mathsf{T}} + \mathbf{s}_t^{\mathsf{T}} \boldsymbol{\theta}_{\text{ATT}} I \tag{9}$$

We first define the typical elements of $\mathbf{s}_t$ and $\boldsymbol{\theta}_{\text{ATT}}$ as below:

$$\mathbf{s}_t = [s_1, s_2, ..., s_N], \tag{10}$$

$$\boldsymbol{\theta}_{\text{ATT}} = [\theta_1, \theta_2, ..., \theta_N], \tag{11}$$

where $N$ denotes the number of elements; $s_i$ and $\theta_i$ are scalar parameters.

$(\boldsymbol{\theta}_{\text{ATT}}^{\mathsf{T}} \mathbf{s}_t)\mathbf{s}_t$ can thus be written as:

$$(\boldsymbol{\theta}_{\text{ATT}}^{\mathsf{T}} \mathbf{s}_t)\mathbf{s}_t = \left(\sum_i \theta_i s_i\right)[s_1, s_2, ..., s_N]. \tag{12}$$

Hence, the Jacobian take the following form:

$$\frac{\partial \mathbf{s}'_t}{\partial \mathbf{s}_t} = \frac{\partial}{\partial \mathbf{s}_t}((\boldsymbol{\theta}^\mathsf{T}_{\mathrm{ATT}}\mathbf{s}_t)\mathbf{s}_t) \tag{13}$$

$$= \begin{bmatrix} \theta_1 s_1 + \sum_i \theta_i s_i & \theta_2 s_1 & ... & \theta_N s_1 \\ \theta_1 s_2 & \theta_2 s_2 + \sum_i \theta_i s_i & ... & \theta_N s_2 \\ \vdots & \vdots & \ddots & \vdots \\ \theta_1 s_N & \theta_2 s_N & ... & \theta_N s_N + \sum_i \theta_i s_i \end{bmatrix} \tag{14}$$

$$= \begin{bmatrix} \theta_1 s_1 & \theta_2 s_1 & ... & \theta_N s_1 \\ \theta_1 s_2 & \theta_2 s_2 & ... & \theta_N s_2 \\ \vdots & \vdots & \ddots & \vdots \\ \theta_1 s_N & \theta_2 s_N & ... & \theta_N s_N \end{bmatrix} + I \sum_i \theta_i s_i \tag{15}$$

$$= \mathbf{s}_t \boldsymbol{\theta}^\mathsf{T}_{\mathrm{ATT}} + \mathbf{s}^\mathsf{T}_t \boldsymbol{\theta}_{\mathrm{ATT}} I, \tag{16}$$

where $I$ is the identity matrix.

## B    DETAILS OF THE MINIRCV1 DATASET

Table 5 contains details of the miniRCV1 single-label and multi-label classification task. The single-label classification task contains 55 classes in total and the multi-label classification task contains 102 labels in total.

Table 5: Number of classes in meta-split of miniRCV1.

|  | Meta-train | Meta-validation | Meta-test |
|---|---|---|---|
| Single-label | 30 | 13 | 12 |
| Multi-label | 70 | 12 | 20 |

## C    ADDITIONAL EMPIRICAL RESULTS

### C.1    THE IMPORTANCE OF ATTENTION

In this section, we include additional empirical results for single-label and multi-label miniRCV1 experiments in Table 6 and Table 7 to show the importance of attention, wherein "meta" denotes the type of meta learner, "Base" denotes the type of classifier, "random" denotes models trained from random initialization, "pretrained" denotes models trained from a pretrained model on the meta-training set, "(A)" denotes models trained with attention and the bold numbers highlight the best performing ones at 95% confidence interval.

The empirical results suggest that attention provides performance improvements regardless of what meta-learner or classifier is used. Given the same meta learning algorithm, adding attention to the classifier always improves model performance.

### C.2    THE IMPACT OF NETWORK ARCHITECTURE

We experimented with both LSTM and TCN as the classifier architecture. Although meta learning works with both LSTM and TCN and they all provide improvements from randomly initialized and pretrained models, it is worthwhile to highlight their different properties. Overall, TCN has faster training speed and generalization when compared with LSTM. One main problem when using LSTM as classifier is that, in meta-training, the LSTM saturates at a very early stage owing to difficulties in optimization, and prevents the meta-learner from obtaining sharable representations across different tasks. Table 8 shows the empirical comparison between bidirectional LSTM and TCN when ATAML is used as the meta learner. The results suggest that TCN performs better than bidirectional LSTM across all experiments on miniReuters-21578.

Table 6: miniRCV1 single-label classification accuracies

| Method | | 5-way Accuracy | | 10-way Accuracy | |
|---|---|---|---|---|---|
| Meta | Base | 1-shot | 5-shot | 1-shot | 5-shot |
| random | TCN | 26.70% | 55.43% | 17.64% | 41.81% |
| random | TCN (A) | 41.52% | 65.64% | 28.32% | 45.12% |
| pretrained | TCN | 22.38% | 37.17% | 10.67% | 27.76% |
| pretrained | TCN (A) | 24.06% | 57.08% | 18.60% | 45.85% |
| MAML | TCN | 33.86% | 61.44% | 22.55% | 41.94% |
| MAML | TCN (A) | 47.09% | **72.65%** | 31.57% | **62.75%** |
| ATAML | TCN (A) | **54.05%** | **72.79%** | **39.48%** | **61.74%** |

Table 7: miniRCV1 multi-label classification

| Method | | 5-way Micro-F1 | | 10-way Micro-F1 | | 5-way Macro-F1 | | 10-way Macro-F1 | |
|---|---|---|---|---|---|---|---|---|---|
| Meta | Base | 1-shot | 5-shot | 1-shot | 5-shot | 1-shot | 5-shot | 1-shot | 5-shot |
| random | TCN | 18.7% | 40.6% | 30.2% | 40.9% | 11.3% | 36.4% | 9.9% | 23.6% |
| random | TCN (A) | 38.9% | 60.9% | 40.6% | 45.6% | 31.4% | 55.7% | 22.8% | 33.1% |
| pretrained | TCN | 25.1% | 36.2% | 28.2% | 35.2% | 17.0% | 30.1% | 9.1% | 20.7% |
| pretrained | TCN (A) | 26.9% | 55.8% | 33.5% | 52.1% | 17.0% | 51.5% | 14.9% | 41.4% |
| MAML | TCN | 35.7% | 45.6% | 20.5% | 40.2% | 22.9% | 41.9% | 7.6% | 27.7% |
| MAML | TCN (A) | 52.3% | **69.1%** | 44.9% | **58.6%** | 43.2% | 64.3% | 27.7% | **48.4%** |
| ATAML | TCN (A) | **59.6%** | **71.1%** | **50.7%** | **61.3%** | **54.3%** | **65.0%** | **38.5%** | **49.2%** |

Table 8: Comparing bidirectional LSTM and TCN as classifier on miniReuters-21578

| Method | | 5-way Micro-F1 | | 10-way Micro-F1 | | 5-way Macro-F1 | | 10-way Macro-F1 | |
|---|---|---|---|---|---|---|---|---|---|
| Meta | Base | 1-shot | 5-shot | 1-shot | 5-shot | 1-shot | 5-shot | 1-shot | 5-shot |
| ATAML | LSTM (A) | 38.0% | 62.3% | 27.1% | 33.7% | 30.3% | 50.2% | 18.8% | 21.2% |
| ATAML | TCN (A) | **59.8%** | **71.1%** | **50.7%** | **61.3%** | **54.3%** | **65.0%** | **38.5%** | **49.2%** |

## C.3 OTHER BASELINE METHODS

Table 9 shows the comparison between the proposed ATAML and classic machine learning methods, i.e., SVM, Naive Bayes Multinomial and KNN, which uses tfidf features as model inputs. The results suggest that SVM and naive Bayes multinomial severely overfit on the training data generalizes poorly on evaluation. The K-nearest neighbor classifier performs better than SVM and naive Bayes multinomial mainly because it is an nonparametric and distance-based algorithm. The proposed ATAML is significantly better than KNN on the Micro-F1 measure and ATAML performs at least as good as KNN on the Macro-F1 measure.

Table 9: Comparing ATAML with SVM, Naive Bayes Multinomial and KNN on miniReuters-21578

| Method | 5-way Micro-F1 | | 10-way Micro-F1 | | 5-way Macro-F1 | | 10-way Macro-F1 | |
|---|---|---|---|---|---|---|---|---|
| | 1-shot | 5-shot | 1-shot | 5-shot | 1-shot | 5-shot | 1-shot | 5-shot |
| SVM | 3.8% | 35.8% | 0.3% | 18.8% | 3.3% | 25.1% | 0.2% | 12.6% |
| Naive Bayes Multinomial | 0.5% | 7.7% | 0.0% | 0.0% | 0.2% | 3.4% | 0.0% | 0.0% |
| KNN | 46.7% | 54.4% | 39.4% | 57.3% | 43.8% | 37.3% | **37.4%** | **52.5%** |
| ATAML, TCN (A) | **59.8%** | **71.1%** | **50.7%** | **61.3%** | **54.3%** | **65.0%** | **38.5%** | **49.2%** |

Table 10 summarizes the comparison between the proposed ATAML and document embedding approaches, i.e., doc2vec (Levine & Haus, 1985) and doc2vecC (Chen, 2017). In contrast to ATAML that uses attention to aggregate information from substructures of some text input, the document embedding approaches directly encode each document into one embedding vector and another classifier, such as KNN (Bailey & Chopra, 2018) or SVM, is applied on the document embeddings for classification.

The empirical results suggest the document embedding approaches are not as effective as the proposed ATAML method. This finding confirms the need to apply attention on substructures of text data, rather than treating each document as a static embedding vector.

Table 10: Comparing ATAML with document embeddings methods on miniReuters-21578

| Method | 5-way Micro-F1 | | 10-way Micro-F1 | | 5-way Macro-F1 | | 10-way Macro-F1 | |
|---|---|---|---|---|---|---|---|---|
| | 1-shot | 5-shot | 1-shot | 5-shot | 1-shot | 5-shot | 1-shot | 5-shot |
| Doc2Vec, KNN | 31.4% | 42.0% | 19.4% | 32.9% | 18.5% | 28.9% | 10.1% | 22.5% |
| Doc2Vec, SVM | 27.4% | 59.1% | 11.4% | 44.3% | 19.9% | 44.6% | 8.5% | 31.0% |
| Doc2VecC, KNN | 42.8% | 62.6% | 30.2% | 50.0% | 34.9% | 53.2% | 23.9% | 42.2% |
| Doc2VecC, SVM | 33.7% | 58.4% | 18.6% | 42.7% | 25.8% | 46.0% | 12.5% | 30.3% |
| ATAML, TCN (A) | **59.6%** | **71.1%** | **50.7%** | **61.3%** | **54.3%** | **65.0%** | **38.5%** | **49.2%** |

