# OpenReview forum: "Attentive Task-Agnostic Meta-Learning for Few-Shot Text Classification"
_ICLR.cc/2019/Conference_

### Official Review · AnonReviewer2 · 2018-10-27
**Attentive Task-Agnostic Meta-Learning for very-few-shot learning**

**Rating:** 7
**Confidence:** 3

**Review:**

The authors introduce the Attentive Task-Agnostic Meta-Learning (ATAML) algorithm for text classification.
The main idea is to learn task-independent representations, while other parameters, including the attention mechanism, are being fine-tuned for each specific task after pretraining.
The authors find that, for few-shot text classification tasks, their proposed approach outperforms several important baselines, e.g., random initialization and MAML, in certain settings. In particular, ATAML performs better than MAML for very few training examples, but in that setting, the gains are significant.

Comments:
- I am unsure if I understand the contributions paragraph, i.e., I cannot count 3 contributions. I further believe the datasets are not a valid contribution, since they are just subsets of the original datasets.
- Using a constant prediction threshold of 0.5 seems unnecessary. Why can't you just tune it?
- 1-shot learning is maybe theoretically interesting, but how relevant is it in practice?

---

### Official Review · AnonReviewer1 · 2018-11-02
**Interesting approach for few-shot text classification**

**Rating:** 5
**Confidence:** 3

**Review:**

This paper presents a meta learning approach for few-shot text classification, where task-specific parameters are used to compute a context-dependent weighted sum of hidden representations for a word sequence and intermediate representations of words are obtained by applying shared model parameters.

The proposed meta learning architecture, namely ATAML, consistently outperforms baselines in terms of 1-shot classification tasks and these results demonstrate that the use of task-specific attention in ATAML has some positive impact on few-shot learning problems. The performance of ATAML on 5-shot classification, by contrast, is similar to its baseline, i.e., MAML. I couldn’t find in the manuscript the reason (or explanation) why the performance gain of ATAML over MAML gets smaller if we provide more examples per class. It would be also interesting to check the performance of both algorithms on 10-shot classification.

This paper has limited its focus on meta learning for few-shot text classification according to the title and experimental setup, but the authors do not properly define the task itself.

---

### Official Review · AnonReviewer3 · 2018-11-05

**Rating:** 5
**Confidence:** 4

**Review:**

Summary of paper: For the few shot text classification task, train a model with MAML where only a subset of parameters (attention parameters in this case) are updated in the inner loop of MAML. The empirical results suggest that this improves over the MAML baseline.

I found this paper confusingly written. The authors hop between a focus on meta-learning to a focus on attention, and it remains unclear to me how these are connected. The description of models is poor -- for example, the ablation mentioned in 4.5.3 is still confusing to me (if the attention parameters are not updated in the inner loop of MAML, then what is?). Furthermore, even basic choices of notation, like A with a bar underneath in a crowded table, seem poorly thought out.

I find the focus on attention a bit bizarre. It's unclear to me how any experiments in the paper suggest that attention is a critical aspect of meta-learning in this model. The TAML baseline (without attention) underperforms the ATAML model (with attention), but all that means is that attention improves representational power, which is not surprising. Why is attention considered an important aspect of meta learning?

To me, the most interesting aspect of this work is the idea of not updating every parameter in the MAML inner loop. So far, I've seen all MAML works update all parameters. The experiments suggest that updating a small subset of parameters can improve results significantly in the 1-shot regime, but the gap between normal MAML and the subset MAML is much smaller in the 5-shot regime. This result suggests updating a subset of parameters can serve as a method to combat overfitting, as the 1-shot regime is much more data constrained than the 5-shot regime.

It's unfortunate that the authors do not dig further down this line of reasoning. When does the gap between MAML on all parameters and only on a subset of parameters become near-zero? Does the choice of the subset of parameters matter? For example, instead of updating the attention weights, what happens if the bottommost weights are updated? How would using pretrained parameters (e.g., language modeling pretraining) in meta-learning affect these results? In general, what can be learned about overfitting in MAML?

To conclude, the paper is not written well and has a distracting focus on attention. While it raises an interesting question about MAML and overfitting, it does not have the experiments needed to explore this topic well.

---

### Meta-Review · Area_Chair1 · 2018-12-13
**Interesting results but very unclear narrative**

**Confidence:** 4
**Recommendation:** Reject

**Metareview:**

This paper describes an incorporation of attention into model agnostic meta learning. The reviewers found that the paper was rather confusing in its presentation of both the method and the tasks. While the results seemed interesting, it was difficult to frame them due to lack of clarity as to what the task is, and the relation between attention and MAML. It sounds like this paper needs a bit more work, and thus is not suitable for publication at this time.

It is disappointing that the reviews were so short, but as the authors did not challenge them, unfortunately the AC must decide on the basis of the first set of comments by reviewers.